# Towards Improved Unmanned Aerial Vehicle Edge Intelligence: A Road Infrastructure Monitoring Case Study

**Sofia Tilon** [1,*] **, Francesco Nex** [1] **, George Vosselman** [1] **, Irene Sevilla de la Llave** [2] **and Norman Kerle** [1]

1   Faculty of Geo-Information Science and Earth Observation (ITC), University of Twente, 7514 AE Enschede, The Netherlands
2   ACCIONA Ingeniería, C. de Anabel Segura, 11, 28108 Alcobendas, Madrid, Spain
*   Correspondence: s.m.tilon@utwente.nl

**Abstract:** Consumer-grade Unmanned Aerial Vehicles (UAVs) are poorly suited to monitor complex scenes where multiple analysis tasks need to be carried out in real-time and in parallel to fulfil time-critical requirements. Therefore, we developed an innovative UAV agnostic system that is able to carry out multiple road infrastructure monitoring tasks simultaneously and in real-time. The aim of the paper is to discuss the system design considerations and the performance of the processing pipeline in terms of computational strain and latency. The system was deployed on a unique typology of UAV and instantiated with realistic placeholder modules that are of importance for infrastructure inspection tasks, such as vehicle detection for traffic monitoring, scene segmentation for qualitative semantic reasoning, and 3D scene reconstruction for large-scale damage detection. The system was validated by carrying out a trial on a highway in Guadalajara, Spain. By utilizing edge computation and remote processing, the end-to-end pipeline, from image capture to information dissemination to drone operators on the ground, takes on average 2.9 s, which is sufficiently quick for road monitoring purposes. The system is dynamic and, therefore, can be extended with additional modules, while continuously accommodating developments in technologies, such as IoT or 5G.

**Keywords:** edge-computation; real-time; remote processing; PANOPTIS; deep learning; DeltaQuad; monitoring; fixed-wing VTOL; hybrid UAV





## 1. Introduction

Unmanned Aerial Vehicles (UAVs) have a prominent place amongst the variety of remote sensing platforms and have been increasingly used in the past decade due to their flexibility and ease of use [1]. Leveraging deep learning, their demonstrated usability for various detection and monitoring tasks in complex scenes has fueled the push for autonomous aerial intelligence with real-time information retrieval at its core [2]. To this end, a growing body of literature has examined the potential of real-time UAV-based systems. This trend is an improvement on traditional UAV-based systems where data analysis tended to be conducted in a post-processing, post-flight fashion.

Real-time UAV-based processing is achieved using edge computation or remote processing technology, where the latter requires high throughput data streaming capacities [3–5]. Both directions show limitations when applied to complicated monitoring scenarios where multiple information needs co-exist in time and space. Edge devices embedded on-board a UAV can work directly with the collected data, and do not require the streaming of data to the remote processing platform. However, they are limited in computational capacities and, therefore, efficient design choices need to be made to reduce computational strain, which oftentimes results in an algorithm that can solve tasks with limited complexity or reduced inference time and performance [3,6,7]. In contrast, remote processing systems have the ability to run intensive computations on hardware that is more powerful than edge devices. However, such systems rely on reliable data throughput to

achieve real-time processing, which can be volatile in practice. Therefore, design choices have to be made that optimize data throughput in order to achieve real-time processing, which often comes at the cost of reduced output quality [8].

Thus far, to the best of our knowledge, no existing UAV-based system has leveraged the advantages of both edge-computation and remote processing frameworks in a single system. Therefore, we designed a UAV agnostic monitoring system that can intelligently balance on-board and remote processing, allowing it to carry out multiple critical monitoring tasks at once, unlike existing systems that focus solely on a single task.

The system was developed in the context of road infrastructure monitoring in the Horizon 2020 PANOPTIS project (www.panoptis.eu accessed on 5 April 2022), which aims at increasing the resilience of road infrastructures by providing road operators with a decision support system for routine and post-disaster scenarios. Road infrastructure operators strive for high road safety standards, and, as such, they are charged with a variety of complex monitoring tasks [9]. The complexity is caused by inherent characteristics of the road infrastructure that relate to the physical dimensions of elements within the road corridor and the nature of processes and events that can occur within it. Long stretches of infrastructure corridors need to be monitored in a semi-continuous fashion for continuous degradation or for sudden manmade or natural events. Because of these characteristics, each monitoring task has different requirements in monitoring or analysis speed and accuracies. Critical events require instantaneous situational awareness, whereas continuous degradation or mapping tasks are less time sensitive. With the past decades showing a continuous rise in traffic flow, traffic loads, and extreme weather events due to climate change, these multi-faceted tasks have become even more complex in terms of resources and execution [10].

In this context, we show that our designed system can carry out different road monitoring tasks, such as on-board object detection or scene segmentation, as well as off-board 3D mapping. All tasks are carried out in the appropriate timespan that matches the nature of the monitoring objective, while placing particular focus on the real-time characteristics of the system. We define a system to be real-time when it can process data immediately once they arrive, and when it can deliver results sufficiently fast to the end-users such that they consider the results instantly available. In practice, this means that the processing pipeline, from image-capture to information dissemination to ground-based personnel, needs to be sufficiently quick that road operators are able to take mitigating actions within seconds. The first version of the system was deployed on a novel typology of UAV, the hybrid UAV, i.e., a fixed-wing Vertical Take Off and Landing (VTOL) platform. It combines the characteristics of VTOL and fixed-wing designs, making it especially suitable for road infrastructure monitoring by being able to survey linear horizontal objects, as well as vertical structures, such as bridges, retaining walls, or surrounding steel structures, whereas regular quadcopter UAVs can typically only carry out a single monitoring task at limited distance ranges. As a result, we present a powerful UAV-based monitoring system that is unique in its ability to achieve multiple monitoring objectives simultaneously and to bring the information to end-users in real-time, whereas existing systems usually only achieve a single task at a time.

This work primarily aims at describing the system in terms of its hardware and software configuration, and, in particular, in terms of its combined edge- and remote-processing pipeline. In line with the importance of designing a system that is capable of carrying out multiple tasks at once, we show that multiple deep learning models can be executed in real-time and in parallel on an embedded edge device. The performance of the implemented deep learning suite in the designed system is discussed in terms of runtime. A detailed discussion of the achieved accuracy or similar metrics by the deep learning modules falls outside the scope of this technical note. Code pertaining to the system has been made publicly available (https://github.com/UAV-Centre-ITC/flightconfig-webapp accessed on 27 January 2022). This paper shows achievements obtained during a validation

trial, which was executed in the context of the PANOPTIS project at the A-2 highway in the Guadalajara province, Spain.

Related work can be found in Section 2; Section 3 describes the technical details of the UAV monitoring system; Section 4 describes benchmark experiments and the test that was executed in Spain; finally, the discussion and conclusion describing the system's limitations and potential future applications can be found in Sections 5 and 6, respectively.

## 2. Related Work

### 2.1. UAV-Based Road Corridor Monitoring

This section begins by examining several applications of UAVs for road infrastructure monitoring tasks, such as degradation detection and obtaining situational awareness in the context of their execution style and hardware requirements. This section ends by summarizing the observed trends in UAV-based infrastructure monitoring and emphasizes the general shortcomings of these frameworks when it comes to monitoring complex scenes. The use of different platforms or sensors for similar tasks, such as mobile mappers, fixed sensors, and laser or hyperspectral sensors, falls outside the scope of this paper and will not be discussed, although we acknowledge their existence and their merits for the road corridor monitoring tasks described in this section.

#### 2.1.1. Degradation Detection

Road operators dedicate a significant portion of monitoring efforts towards the inspection of road surfaces because they serve, not only as a proxy for structural health, but can also influence the degree of driver-induced incidents [11]. Road surface inspection using UAVs is achieved by deriving geometric features from photogrammetry products. Several examples can be found in literature. Zhang and Elaksher [12] developed algorithms for UAV image orientation and 3D surface reconstruction to map unpaved road surface conditions and achieved sub-centimeter accuracies. Nappo et al. [13] used a combination of 3D reconstruction and orthoimage products to derive qualitative road condition parameters, such as the International Roughness Index (IRI), for paved roads affected by a slow moving landslide. Tan and Li [14], Roberts et al. [15], and Biçici and Zeybek [16] all demonstrated with slightly varying approaches how to fit a photogrammetry-derived dense point cloud with a plane to extract road damage features. Saad and Tahar [17] showed that potholes and rutting could be extracted by using commercial 3D reconstruction and road damage analysis software using a VTOL UAV.

The inspection of bridges or steel structures is another major aspect of road infrastructure monitoring [18–21]. A majority of western infrastructure assets that were erected during the construction boom of the 1960s will soon reach the end of their design life and, therefore, require major investments in the upcoming decade to keep them safe for road users [22]. Oftentimes, the main objective is to find cracks, spalling, or corrosion. UAVs are used to substitute or complement unsafe and expensive manual monitoring efforts, although the implementation of a successful inspection program is not trivial [18]. Various efforts towards this aim have been published. Humpe [19] demonstrated the usability off an off-the-shelf UAV and a 360-degree camera for crack detection, Morgenthal and Hallerman [20] evaluated factors influencing image quality to achieve accurate structure damage assessment and Chen et al. [21] used 3D reconstruction to monitor a bridge for volume loss.

Degradation detection is a continuous non-urgent task and, therefore, the works mentioned above are all executed in a post-processing fashion, where data are transferred from the UAV to a workstation post-flight to carry out the analysis. This method is justified considering the computational strain of geometrical feature derivation or creating photogrammetry products and the urgency of the task.

### 2.1.2. Situational Awareness

Another important task for road operators is obtaining situational awareness of their road corridor in a semi-continuous fashion. These tasks include the detection of anomalous events, such as vehicle accident detection or the detection of foreign objects (debris, animals). Because these scenarios affect driver's safety and traffic flow these tasks are extremely time-sensitive and, therefore, real-time and edge computation are already commonly used in studies that address these tasks. Nguyen et al. [23] achieved accurate vehicle object detection using an edge device by optimizing the pre-trained detection model for usage on the edge device. Li et al. [24] developed a real-time remote processing framework to detect, track, and estimate the speed of vehicles. Similarly, Balamuralidhar et al. [4] developed a vehicle detection, vehicle speed, and vehicle tracking framework that was optimized to work on-board a UAV in real-time, yielding high accuracies.

Special cases of anomalous events that are not exclusive to road corridors but could have negative consequences nonetheless, are hazardous events, such as floods, landslides, or wildfires. With an increase of climate change-induced extreme weather events, these tasks have become a vital part when monitoring sensitive structures. The usage of UAVs has been explored in the context of flood detection or road surface damage detection in landslide-affected areas [13,25,26]. In addition, a large body of work has demonstrated the value of UAVs for structural damage detection after earthquakes [27]. Still, few examples can be found for real-time UAV-based hazard detection, with the exception of the work by Jiao et al. [28] who implemented a deep learning model for on-board forest fire detection, and more recently, the work by Hernández et al. [25] who presented a model for real-time flooded area detection.

### 2.1.3. Scene Understanding

Finally, despite the growing recognition of the need of semantic labels to enable the execution of other time-sensitive tasks and the growing interest in the development of low latency semantic segmentation models that can be deployed on edge devices, it is observed that road corridor scene understanding is mostly overlooked [29]. From the monitoring and maintenance perspective, knowledge of the shape or edge of road markings, road surface damage, or the road surface itself allows road operators to obtain qualitative information on the state of the infrastructure using semantic reasoning. For example, Chen and Dou [30] showed that semantic labels aided the detection of road obstructions and road surface damages after earthquake events. However, road scene understanding is mostly approached from an autonomous driving perspective and rarely from a road corridor monitoring perspective, let alone a UAV-based perspective [31–33]. An exception is the work of Yang et al. [34], who developed a lightweight Context Aggregation Network that achieved high accuracy scores on UAV segmentation benchmark datasets with high execution speeds on an edge device. Nonetheless, the datasets that were used are not refined to contain class labels that matter to road operators, such as road markings or other road furniture classes [35,36]. Therefore, the usability of this model for infrastructure monitoring purposes remains to be seen.

The works described above show that various monitoring objectives co-exist in complex scenes such as road corridors. Moreover, for road infrastructures in particular, there is a clear distinction between urgent (situational awareness) and non-urgent (degradation detection) tasks, which are characteristically carried out in either a post-processing or an edge computational fashion. However, it can be reasonably assumed that road operators would be interested in achieving several urgent and non-urgent objectives at once. Nonetheless, each of the works described above address solely single objectives to which the data acquisition and the post-processing pipeline is tailored. We argue that a UAV-based system that can address multiple objectives at once by smartly combining edge processing and remote processing can make UAV-based monitoring efforts more efficient and practical.

*2.2. Real-Time UAV Monitoring Systems*

This section discusses different real-time UAV-based monitoring systems that also consider the transmission of information to a ground station in real-time and examines what can be learned from them. This section does not describe real-time flight path planning or optimization, because our system constraints the UAV mission within the responsibilities of the UAV pilot for safety and regulation motivations.

Regarding the use of commercial products, companies such as DroneDeploy, DataFrom-Sky, Sunflower Labs and others offer real-time object detection for security and monitoring purposes. While these products are characterized by high-quality customer interfaces and support, their proprietary nature is the main drawback. Moreover, they are often designed for specific UAV models and single tasks, making them uninteresting for the more complex monitoring scenes.

Most non-commercial studies use data compressions to decrease transmission rates, such as image compression or decreasing image resolution. For example, Hein et al. [8] designed a security UAV system to generate real-time high-resolution maps. The authors correctly considered that pre-screening data on-board allows for the transmission of solely those images that contain vital information, thus preserving bandwidth. Their system yielded high data throughput and quality over the course of several experiments, by applying data compressions and Terrain Aware Clipping techniques. Similarly, Meng et al. [37] developed a real-time excavator detection system to prevent damages to pipelines. They deployed a You Only Look Once (YOLO) v3 model on a UAV-embedded edge device, and transmitted compressed images and warning messages to a ground control station with an impressive maximum time of 1.15 s. The image resolution was chosen to be as low as possible ($288 \times 288$) without affecting the detection accuracy. Choosing data compressions is valid for a single task system. However, when an image is used for multiple tasks, such as in this study, fixed image resolutions are not desirable. Moreover, greater image resolutions are expected to achieve better results for a majority of tasks, including photogrammetry and deep learning.

Besides the single objective frameworks mentioned above, similar to our approach, some studies aimed at using an edge-enabled UAV platform in a holistic monitoring approach. In addition to data compressions, other factors such as network and legacy systems were found to play a role. Hossain and Lee [38] conducted a comprehensive study of the performance differences between (non-)graphic processing unit (GPU)-embedded devices and GPU enabled ground control stations (GCS) to support real-time information dissemination. They used a socket protocol to transmit data to the GCS on the condition that both devices were running on the same network. Although minor lag was reported, it remains unclear what transmission rates were achieved. Nonetheless, most monitoring systems, including the one presented here, cannot rely on multiple devices existing in the same network, because network coverage can be sparse or of low quality, especially in remote road infrastructure corridors. In another holistic approach, Maltezos et al. [39] designed a comprehensive real-time system to create situational awareness where a UAV leveraged information from terrestrial monitoring assets and transmitted the information produced on-board to third party workstations. In this manner, multiple tasks could be carried out on different devices within the platform. Although the authors mentioned necessary video compressions to transmit video over an Internet Protocol (IP) network interface, no detail was given on the resulting latency of the system. In conclusion, real-time communication between the UAV and remote (legacy) devices requires additional considerations.

The systems described above show that a multitude of design choices can be made when it comes to data transmissions, resource allocation, or choice of deep learning model. However, these are only some of the many challenges that need to be considered when designing a real-time UAV system [40]. In an excellent review by Yazid et al. [40] it is stated that the mixing fields of Internet of Things (IoT) and mobile edge computation for UAVs require attention to be given to interference, interoperability, security, or scalability, to name a few. Ejaz et al. [41] emphasized that efficient connectivity is one of the main challenges

when designing an IoT UAV system, confirmed by Meng et al. [37], who found that 4G coverage was the main bottleneck affecting transmission speed. Ejaz et al. [41] described the multitude of design choices that can be made and designed a named data networking (NDN) IoT 4G-sensor network to supply end-users with real-time wildfire notifications, coming from infrared flame and smoke sensors on-board a UAV. In complementary work, Mignardi et al. [42] investigated how a narrowband IoT network for data transmissions could influence energy consumption or latency, which is important for systems with finite energy. IoT developments are ongoing, including promising developments in 5G technology that will surely influence real-time UAV developments. A guide towards 5G-geared design choices for UAV systems can be found in Zeng et al. [43].

## 3. Real-Time UAV Monitoring System

Figure 1 provides a simplified overview of our proposed UAV monitoring system. It consists of three hardware modules, the UAV, its controller, and a GCS. The UAV is equipped with an edge computation device that is capable of executing pre-trained deep learning models. This "deep learning suite" produces information that is needed to create situational awareness for urgent scenarios. Relevant information is transmitted to the GCS in real-time. The GCS hosts a web application. This application is used pre-flight to communicate with the edge device on-board the UAV and to configure the deep learning suite remotely. During the mission, the relevant situational awareness information received from the UAV is displayed inside the web application so that the road operators can review them instantly. Finally, the GCS hosts the "analytical suite". It processes a video stream originating from the UAV and produces information that is needed for degradation detection for non-urgent mitigation measures.

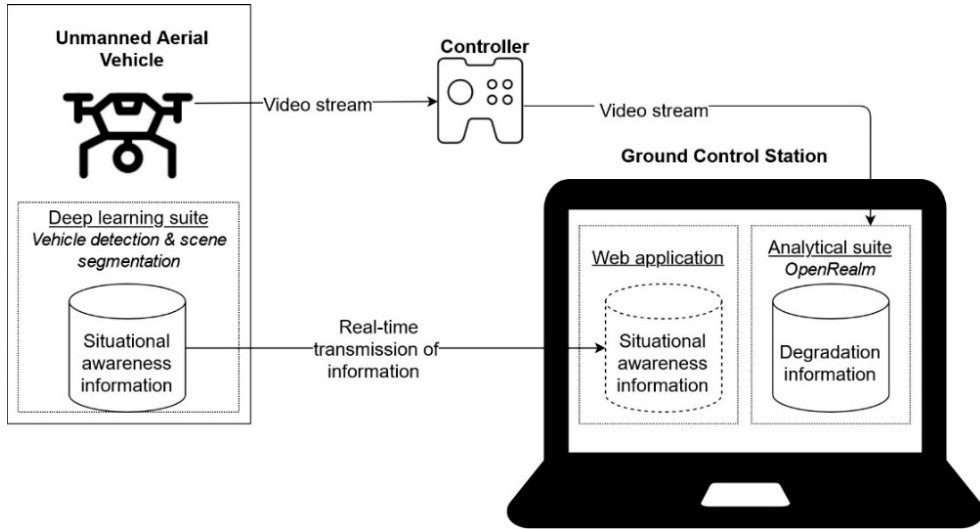

**Figure 1.** Method overview of the designed UAV monitoring system showing where each module is situated and how the processing pipeline functions during a UAV mission.

The rest of this section elaborates on this short overview. Section 3.1 explains the system design criteria. Section 3.2 details the technical specifications of the UAV that was used for deployment. Section 3.3 describes the hardware and software architecture, the communication protocols, and the information flows. Sections 3.4–3.6 describe the analytical suite, deep learning suite, and the web application is succinct detail.

### 3.1. System Design Considerations

As explained above, the UAV monitoring system should be able to be deployed in scenarios where various tasks need to be carried out in varying execution times. Therefore, the system should adhere to the following criteria: it should be able to (i) carry out several

monitoring objectives at once, (ii) optimize the usage of hardware, and (iii) transmit time-critical monitoring information to road operators in real-time. The first criterion and the development of this work in the context of road infrastructure resulted in the system to be designed around three monitoring objectives that are well known to road operators: vehicle detection, scene segmentation, and 3D map generation. These objectives are placeholders within the system and can be replaced by objectives that are relevant for other complex scenes, such as rail infrastructures or industry complexes. The second criterion dictated that the system should consider which hardware component the tasks could be most optimally executed on. Tasks were manually assigned to either the edge device or the GCS. No task scheduling or task-offloading models were used to keep the approach simple. The second criterion dictated that 3D map generation should be executed off-board because it is a computationally heavy task. In the same vein, the second criterion dictated that vehicle detection and image segmentation should be executed on the embedded edge-device to ensure local and fast information generation. Finally, the third criterion imposed that the vehicle detection and image segmentation modules should have low inference times, and that transmission speeds to the GCS should be within seconds. This meant that particular attention was placed first on picking fast inference deep learning models with few parameters to reduce its storage size on the edge device and second on constructing a transmission pipeline that resulted in the least amount of latency.

### 3.2. Fixed-Wing VTOL UAV: The DeltaQuad Pro

The designed hardware and software architecture is UAV agnostic (see Section 3.3). However, different typologies of UAVs suit different monitoring tasks and, therefore, a system was deployed on a UAV that is particularly suited for complex scenes, such as road corridors. A conventional VTOL UAV can approach an object up-close and, with a sensor positioned obliquely, is fit to inspect bridges or other vertical assets. A fixed-wing UAV can fly for long periods in a linear direction and, with a sensor positioned in nadir, is fit to inspect road surfaces or adjacent areas. A fixed-wing VTOL combines these typologies and, therefore, can both fly long distances and hover close to objects. Consequently, a broad range of monitoring tasks can be achieved using a single platform. In the infrastructure context, this means that both road surface inspection and vertical asset (bridges, steel structures) inspection can be achieved using a single platform. The system was therefore deployed on Vertical Technologies' VTOL UAV, the DeltaQuad Pro (Figure 2) but can be deployed on other typologies of drones. Its specifications are listed in Table 1.

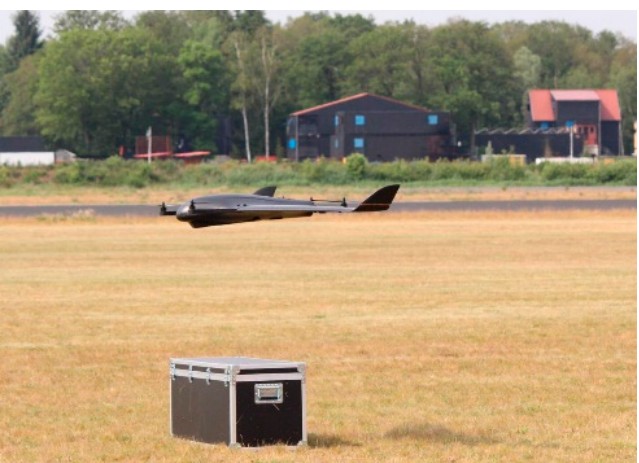 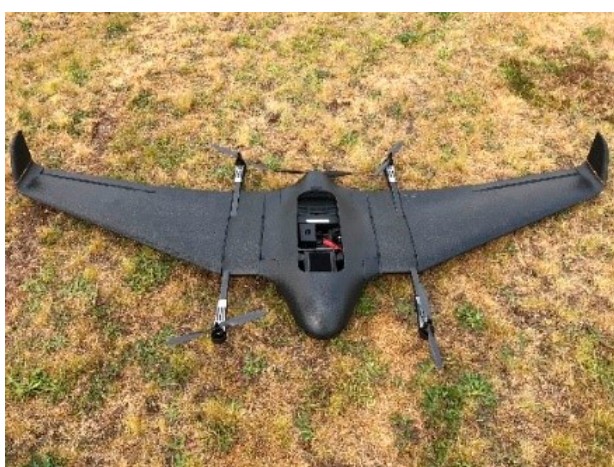

**Figure 2.** Vertical Technologies' DeltaQuad Pro (www.deltaquad.com accessed on 28 January 2022).

**Table 1.** Technical specifications for the DeltaQuad Pro (n.a. = not applicable).

| Characteristic | Value |
| --- | --- |
| Type UAV | Hybrid (Fixed-wing VTOL) |
| Wingspan | 235 cm |
| Height | n.a. |
| Camera | S.O.D.A. (20 Megapixel) |
| Processing unit | NVIDIA Jetson TX2 |
| Long Term Evolution (LTE) | 4G LTE dongle |
| Min–Max speed | 12–28 m/s (fixed-wing) |
| Maximum flight time | 2 h (fixed-wing) |
| Maximum Take of Weight (MTOW) | 6.2 kg |
| Maximum wind speed | 33 km/h |
| Weather | drizzle |
| Autopilot | Px4 Professional autopilot |
| Communication protocol | MAVLink |
| Mission planner | QGroundControl |
| Safety protocol | PX4 safety operations |

### *3.3. Hardware Ecosystem*

This section describes the hardware and software architecture, the communication protocols and the information flows based on the detailed ecosystem overview shown in Figure 3.

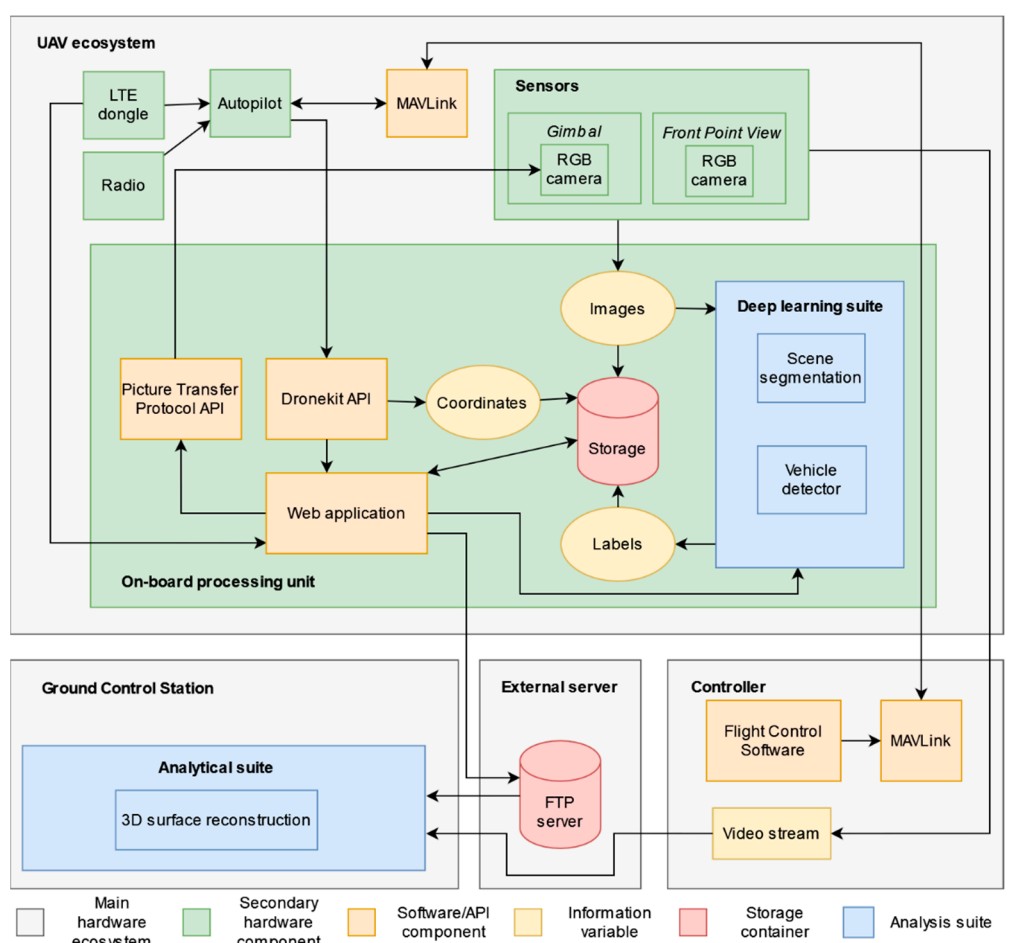

**Figure 3.** Detailed overview of the designed monitoring system showing the communication flow and protocol of hardware components and information.

The UAV controller is the link to the UAV, from which the UAV pilot can carry out the mission and take intervening safety measures when necessary. In addition, it can be used to create mission plans and upload them to the UAV using the 2.4 GHz radio communication link (with a transmission range that can reach up to 30 km) or using the MAVLink communication protocol. When connected to a 5 GHz Long Term Evolution (LTE) hotspot, the controller can download up-to-date satellite maps or connect to the UAV over 4G LTE. The controller displays the video feed from either the Front View Point (FPV) camera or the nadir camera that is placed on a gimbal. The angle of the gimbal with respects to the nadir-line is controlled by a servomotor that can be controlled by a switch on the controller, allowing the user to adjust the field of view and observe scenes in oblique. This is especially relevant considering how safety regulations typically dictate that UAVs need to fly alongside roads, instead of directly above them, consequently rendering the FPV camera unusable and the nadir camera a necessity for monitoring road corridors. While images are forwarded to the GCS via a File Transfer Protocol (FTP)-server, the video stream is directly forwarded to the analytical suite on the GCS, both using 4G LTE connection.

The GCS is a laptop with a NVIDIA GeForce RTX 3080 GPU, an eight core 2.40 GHz central processing unit (CPU), and 32 GB random access memory (RAM), making it a powerful analytical station that can be used in the field. It is used to run the analytical suite that contains modules that cannot or are not urgent enough to be run on the on-board processing unit. As stated earlier, the analytical suite is modular and scenario specific, meaning that it can contain the modules pertaining to the scenario at hand. In this case, the analytical suite contained a real-time 3D surface reconstruction module, based on OpenREALM [44]. Finally, the GCS enables the UAV pilot or secondary users to configure the processing that needs to be executed on-board and to monitor the obtained information in real-time from a web browser interface running from a web application, which is hosted on the UAVs on-board unit.

The UAVs ecosystem consists of an on-board processing unit, an LTE dongle, a radio telemetry module, an autopilot module, and RGB camera units. The processing unit hosts the deep learning suite and the web application. The UAVs internal imaging pipeline during a flight mission has several stages. It starts when the downward facing (S.O.D.A.) camera is triggered to capture images using the Picture Transfer Protocol (PTP) API. In parallel, Global Navigation Satellite System (GNSS) information is retrieved and stamped into the image EXIF metadata using the Dronekit API. The images and other flight data are internally organized in dedicated flight directories. The images are stored in their respective flight project folder, from where the deep learning suite can retrieve them instantaneously once they appear. The deep learning suite runs asynchronously to the image capture pipeline. The information produced by the deep learning suite, such as annotated images with bounding boxes, segmented images, or textual information, are stored within their respective directories. If instructed by the user to do so, the information is asynchronously transferred to the FTP server, from which the GCS retrieves them in real-time by using the FTP mirroring protocol. By running every stage of the processing pipeline asynchronously, none of them is dependent on other stages to finish, aiding fast processing and information dissemination.

### 3.4. Analytical Suite

As explained earlier, the analytical suite runs on the GCS and consists of those modules that are typically computationally heavy and, therefore, cannot be executed on an edge device or should not be executed on the edge device because they are not urgent. The suite is modular and can be extended to include any module that is required for the scene at hand. Here, the analytical suite consists of 3D surface reconstruction using OpenREALM [44]. The original OpenREALM was adapted to process video streams (full-resolution video frames) instead of images. The video stream is forwarded over a 5 GHz LTE connection. No further adaptations were made. Technical details on OpenREALM can be found in the original paper [44].

### 3.5. Deep Learning Suite

The deep learning suite is deployed on the UAVs on-board unit such that images can be processed close to the sensor in real-time. Just like the analytical suite, the deep learning suite is modular and can be adapted to the end-users needs. In this study, it contains modules for vehicle detection to achieve traffic monitoring and for scene segmentation to achieve semantic reasoning for scene understanding. Keeping the third system design consideration in mind (Section 3.1), the modules should be low-weight and -inference. Therefore, the vehicle detection module originates from the MultEYE system, which consist of a lightweight model designed specifically to function optimally on on-board devices during inference [4]. The scene segmentation module is based on the Context Aggregation Network designed for fast inference on embedded devices [34]. The pre-trained segmentation network was fine-tuned for the classes Asphalt, Road markings, Other Infrastructure (including guardrails, traffic signs, steel signposts, etc.), Vehicle, and Truck.

Every module in the deep learning suite was found to benefit from a warm-up period pre-flight to allow for accurate inference time measurements. Within this period, the GPU and CPU of the edge-device are initialized by loading the respective deep learning modules and the corresponding weights and biases into active memory and by carrying out inference on a dummy-image. Inferences on subsequent images now purely reflect inference time only. The inference time for vehicle detection using a dummy image is on average 16 s and drops to 0.33 s on the first subsequent image. The warm-up takes place before the UAV mission launch and, therefore, does not hinder real-time information transmission.

### 3.6. Web Application

A web application was designed to allow a road operator to configure remotely edge-device processing parameters, and to view information produced by the deep learning suite mid-flight. Other parameters, such as flight path configurations, are not handled by the web application but by traditional means of using flight control software operated through the controller or the GCS. This ensures that the edge device and other processes have no influence on the UAV's safety features. The Django-based (www.djangoproject.com accessed on 24 February 2022) web application could be exposed to remote devices such as the GCS, by either hosting it on a public web server or by hosting it on-board the edge device. Both design options had insignificant influence on demonstrating the viability of the designed system and, therefore, the simpler approach was chosen by exposing it on the edge device using a secure http tunneling framework called ngrok (www.ngrok.com accessed on 24 February 2022), while leveraging network connectivity through 4G LTE. The code pertaining to this section has been made available (https://github.com/UAV-Centre-ITC/flightconfig-webapp accessed on 27 January 2022).

Figure 4a shows the landing page where users can set mission parameters that pertain to processes that need to be carried out mid-flight and on-board the edge-device:

(1)　The flight organization ("Flightname");
(2)　The deep learning suite, e.g., which models to execute ("Model");
(3)　The on-board camera and its associated capturing protocol ("Camera");
(4)　The name of the GCS ("Groundstation");
(5)　Which information variables should be transmitted mid-flight ("Transfer");
(6)　Whether these variables should be compressed ("Compressed"); or
(7)　Ancillary information ("Notes").

Figure 4b shows the monitoring dashboard that appears once the flight mission is started. Its purpose is to depict relevant information that is produced on the edge-device, so that the road operator can monitor the flight in real-time. The "deep learning log" shows information, such as number of objects and object labels found in an image, produced by the deep learning suite mid-flight instantaneously once the information appears. The "transfer log" shows which of the selected information variables have been transferred to the FTP-server once the transfer is finished. In addition, to allow the road operators to

confirm or further interpret the information retrieved by the deep learning suite, thumbnail images of the original, segmented, and object detection models are depicted.

**Figure 4.** Web application front-end. (**a**) The landing page where users configure the UAV mission. (**b**) The monitoring dashboard showing mid-flight transfer logs, information variables produced by the deep learning suite, and image thumbnails for visualization purposes.

## 4. Experiment and Benchmarks

The system deployed on the DeltaQuad Pro was validated by means of a trial carried out at a highway. Afterwards, using the collected data, benchmark tests were executed to identify the systems real-time characteristics and potential bottlenecks.

### 4.1. Demo: Road Infrastructure Scenario

The A-2 highway in Guadalajara, Spain, is managed by Acciona Concesiones S.A. PANOPTIS trial activities took place over a section of 77.5 km, and the UAV trial took place at kilometer point 83. This location was chosen based on the absence of sensitive assets, such as power lines, petrol stations, and communication towers, and the presence of mobile network coverage. Another favorable aspect to this location was the presence of a non-public service road, running parallel to the main highway, where anomalous road infrastructure scenes could be simulated. Two vehicles simulating a traffic accident and one foreign object (single safety barrier) simulating debris, were placed on this road. Figure 5a shows UAV images of the scene. The main goal of the trial was to run the deep learning suite and the analytical suite modules simultaneously and to record the performance of the designed system in a realistic scenario from a speed and functional perspective. The flight was configured to transfer only full resolution images to the GCS. The flight was conducted under the European Union Aviation Safety Agency's (EASA) Standard Scenario (STS), specifically, the STS-02 for Beyond Visual Line Of Sight (BVLOS) with airspace observers over a controlled ground surface in a sparsely populated area. This STS dictated that the

UAV flightpath should stay within 50 and 60 m altitude and at 15 m distance from the main A-2 highway. Moreover, the UAV could fly a distance of maximum 2 km from take-off and an observer needed to be positioned at the 1 km interval. In total, a single mission was repeated five times. Each flight took on average 9 min, and 1132 images were collected in total.

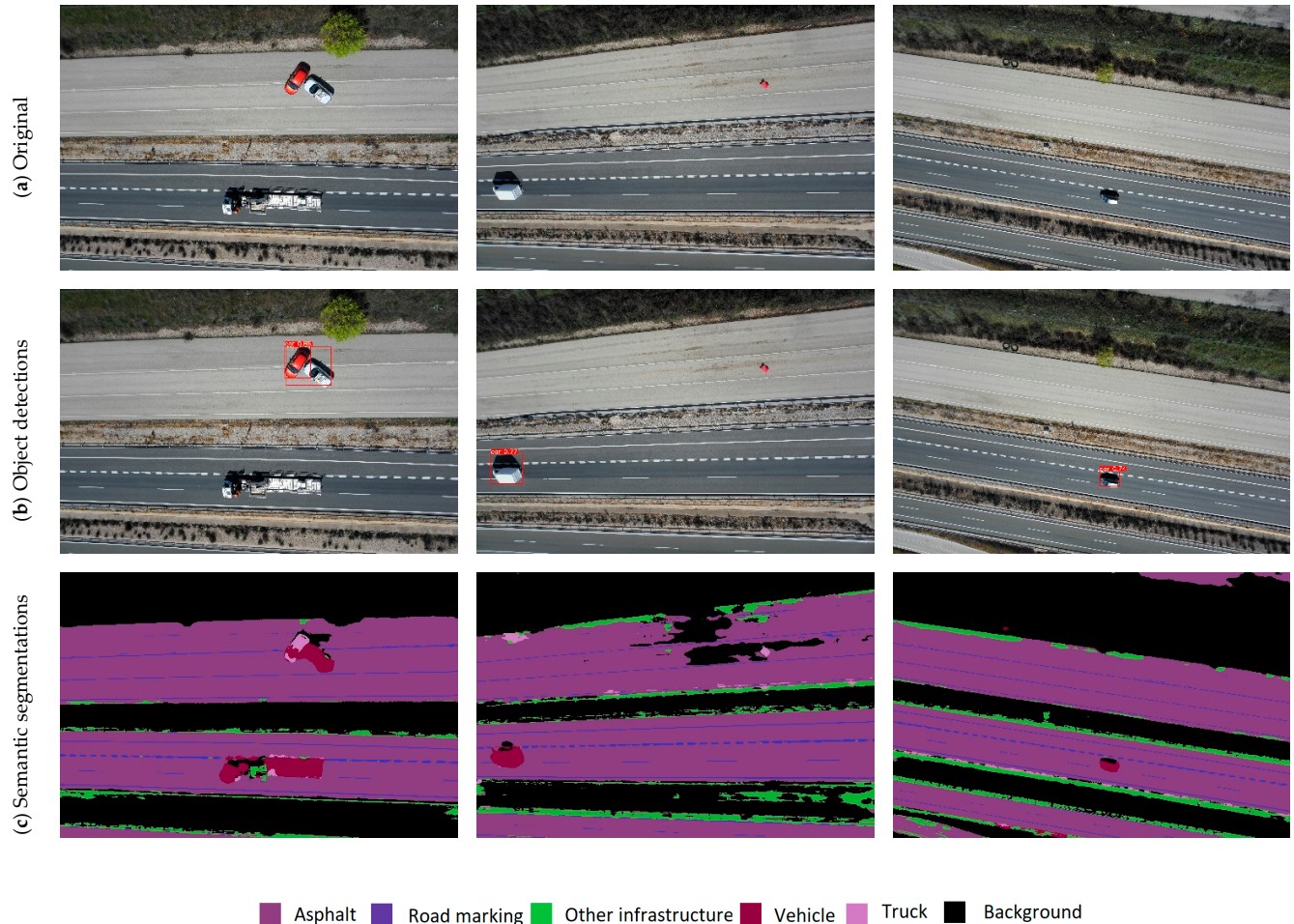

**Figure 5.** (**a**) Images from the trial site showing two lanes of the main A-2 highway (**bottom**) and the service road (**top**) where a traffic incident and foreign object scenario were simulated. (**b**) Results from the vehicle detector module. (**c**) Results from the scene segmentation module.

Figure 5b,c shows examples of information produced by the vehicle detector and scene segmentation modules. Only images and image labels corresponding to those that were found to contain vehicles were transmitted to the GCS to reduce the amount of data that needed to be transferred. This way, for one of the six mission that captured 127 images in total, only 45 images and labels needed to be transferred to the ground, reducing the bandwidth and time needed to obtain relevant information to the ground operator in real-time. The average transfer times are discussed in Section 4.2. The transmission of a single image resized to a smaller resolution and the corresponding label was received by the GCS almost instantaneously (<1 s). Finally, the analytical suite on the GCS carried out OpenREALM without lag or latency.

This trial showed that the hardware and real-time communication pipeline was functioning as expected. Figure 4b depicts a screenshot of the monitoring dashboard taken during the trial and showed that it was able to deliver information to the ground operator mid-flight. The following section will discuss the speed of the system and other performance indicators.

*4.2. Benchmarks*

In order to identify the systems real-time characteristics and potential bottlenecks, several benchmark tests were executed with different user-defined settings and while using the images collected during the trial. These tests shed light on the systems performance in terms of latency and computational strain on the hardware and provided best practice insights.

4.2.1. Latency

Table 2 shows the benchmark results of the system in terms of transfer times and time needed to complete various stages on the on-board processing unit. The average image size is 4.35 Mb for an image resolution of 5472 × 3648 pixels. When limited by network connectivity or coverage, the user has the option to transfer images that are resized without cropping using linear interpolation to a resolution of 240 × 240 pixels, resulting in an average image size of 2.90 Kb. For now, images are now mainly transferred to the GCS for visualization purposes and not for secondary processing. Therefore, when images are not necessary the user has the option to send only the labels (.json files) created by the deep learning suite, which have an average size of 1 Kb (Table 2; column 2). These .json files provide bounding box coordinates for objects found in the image (For example: {"SODA0159.JPG": {"car": [[4161, 1314, 5201, 2465], . . . ]}}). The average download time (Table 2; column 3) refers to the time it takes for the PTP-protocol to trigger the capture signal on the S.O.D.A. camera, and to download the captured image to the on-board processing storage device. This process takes on average 1.09 s. The total average inference time of the vehicle detector and scene segmentation model is 0.343 s per image (Table 2; column 4), which is significantly fast despite their strain on the on-boards GPU engine. Finally, transmission times were measured, which is the time it takes to transfer an image from the UAV's internal storage to the FTP-server. Although the presence of WiFi is unlikely in the field, we measured transfer times over WiFi as the baseline for a full resolution image. Transferring a full resolution image over WiFi took 1.496 s on average (Table 2; column 5). In contrast, transferring full resolution images over 4G took on average 6.043 s (Table 2; column 6). Better results were achieved when transferring resized or labels only with 1.557 and 1.343 s, respectively. These alternative pipelines are deemed acceptable, considering how the compressed images are only needed for visualization purpose, the deep learning suite is executed on-board, and the analytical suite functions using the video stream. The final chain in the pipeline is the transmission of data from the FTP-server to the GCS, which allows road operators to inspect and verify the data whilst in the field. By mirroring the FTP-server every second to the GCS, almost instantaneous (<1 s) data transmission was observed over 4G. In addition, the video-stream was received from the UAVs controller without obvious lags.

**Table 2.** Average inference and transfer times for various sized information objects over 4G LTE and WiFi.

| Item | Size [Mb] | Avg. Download Time [s] | Avg. Inference Time [s] | Avg. Transfer Time over WiFi [s] | Avg. Transfer Time over 4G [s] |
|------|-----------|------------------------|-------------------------|----------------------------------|--------------------------------|
| Original | 4.35 | 1.09 | 0.343 | 1.496 | 6.043 |
| Compressed | 0.0029 | - | - | - | 1.557 |
| Labels | 0.0010 | - | - | - | 1.343 |

From these results, similar to findings by Meng et al. [37], it is concluded that the 4G data plan is the main bottleneck of the designed system, making it difficult to transfer full-resolution imaged over LTE without long transmission times. Therefore, the most optimal pipeline comprises sending compressed images or labels only to the GCS without the loss of vital information. The end-to-end pipeline, from image capture to information dissemination to drone operators on the ground, takes on average (1.09 s + 0.343 s + 1.557 s) 2.99 s for compressed images or (1.09 s + 0.343 s + 1.343 s) 2.78 s for labels only.

These results reveal two things. First, the transmission speed of relevant information, i.e., vehicle locations, to road operators is sufficiently quick to qualify as real-time within the road monitoring context. Second, the real-time transmission of (full-sized) imagery data, which is most desirable as explained in Section 2.2, is non-trivial. Improvements to the transmission pipeline would be required if the system were to be applied to cases that require full-sized imagery for secondary processing, such as photogrammetry. However, in this case, 3D reconstruction is addressed using OpenREALM and video streams, which bypasses this need.

In summary, this test showed that the system is capable of addressing both urgent and non-urgent objectives and that in practice a human operator is required to weigh the need for fast transmission speeds or high data resolutions.

### 4.2.2. Computational Strain

Finally, the performance of the edge device was inspected in order to identify potential bottlenecks influencing inference times. A CPU or GPU load at 100% indicates that the device is trying to process more than it can handle, leading to lagging processing times. Figure 6 shows the loads that the edge device experiences with and without running the deep learning suite. With deep learning, the overall loads increased as expected. Both MultEYE and the Context Aggregation Network were designed specifically to perform optimally on edge devices. It was observed, however, that the GPU and CPU regularly reach their 100% load capacity, risking potential lag. Nevertheless, the pipeline was never blocked or terminated, meaning that the edge device was sufficiently capable of carrying out multiple deep learning processes at the same time.

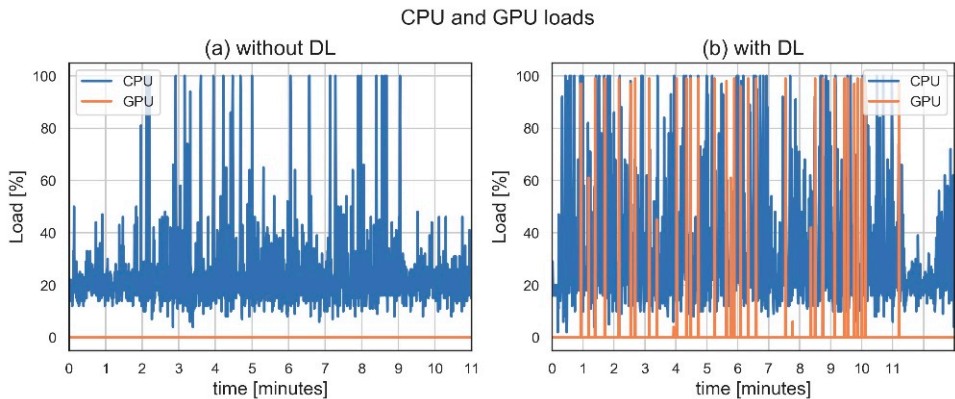

**Figure 6.** CPU and GPU loads on the edge device without (**a**) and with (**b**) the deep learning (DL) suite.

A high power usage of the edge device influences the battery capacity of the UAV system, reducing the maximum flight time. While executing deep learning, a point of concern could be the production of high power surges that lead to unpredictable flight safety performances. Therefore, the edge devices power consumption was investigated while reaching a maximum computational strain. NVIDIA devices can generally operate at either 10 or 15 W; however, a power profile of 10 W is mandatory in order to preserve power for other vital DeltaQuad components. Figure 7 shows an excerpt of CPU and GPU loads, as displayed in Figure 6b, and compares them with power usage. It shows that the cumulative power usage is well below the average power threshold of 10 W. These results show that a lower operating state can be safely chosen, without causing significant bottlenecks while running the deep learning suite. Finally, the GCS was observed to run the OpenREALM 3D surface reconstruction without major bottlenecks or while reaching maximum CPU, RAM, or GPU strains. The maximum CPU strain while executing OpenREALM is 35% per core (280% over eight cores).

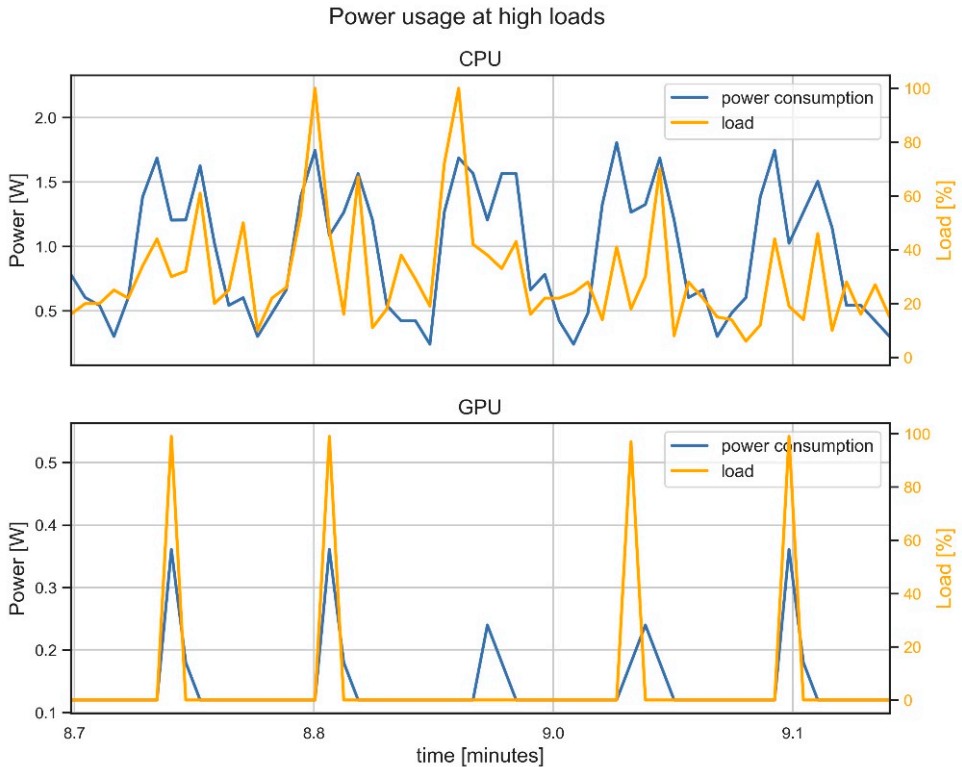

**Figure 7.** Power usage of the on-board unit at high computational CPU and GPU loads.

## 5. Discussion

First, potential improvements to the systems' communication and processing pipeline are discussed. Second, operational constraints are addressed. Finally, the contribution of this system to the field of remote sensing is argued.

Regarding the communication pipeline, the web-application design and deployment and LTE connectivity should be considered for improvements. The web application was designed using Django because it is well documented and easy to use. However, other lightweight options, such as OneNet or Python Socket Server, exist and could potentially facilitate a faster communication pipeline. Moreover, although deploying the web application on the edge device can be considered unusual, this choice was made because public deployment comes with security and privacy considerations that need to be scrutinized, especially when operating the UAV alongside sensitive structures, such as road infrastructures or industrial sites. However, local deployment limits the amount of traffic to the web server and puts more strain on the edge device. Although the system for now could function in a constrained and local setting, with careful consideration future work might consider deploying the web application on a scalable web server, allowing multiple road operators to view the same results at once. Finally, it is unknown to what extent data transmissions are affected by 4G network coverage. Future work will investigate this aspect while also assessing the performance of the system on a 5G network.

Regarding the processing pipeline, as stated in the introduction, the aim of this study was not to optimize the deep learning or analytical suite. Nonetheless, obvious improvements could be made to improve their execution pipeline or to increase their output performance and inference speed. Starting with the latter, increased output performance and inference speeds could be achieved by using different deep learning architectures, by increasing the number of detailed training samples or by optimizing practical flight parameters, such as flying height, to improve the input image quality. Of greater relevance to our aim is the execution pipeline. As was illustrated in Section 2.2, a multitude of choices can be made for various stages within this pipeline. Future work might look into the direction of solutions that are dedicated to achieving steadier load distributions over

the available resources of the edge device, such as NVIDIA TensorRT or Robot Operating System (ROS).

From an operational point of view, regulations pertaining to flying safety and privacy should be acknowledged. Testing and deploying a system that aims at monitoring complex and sensitive scenes is inherently difficult and requires specific flight mission adaptations to adhere to them, such as keeping a certain distance to the road and road users. The trial conducted at the A-2 highway in Spain presented a unique opportunity to test the system in a real world scenario within the boundaries of these regulations. More intensive testing at similar locations is required to improve continuously on the system.

The system presented here is one of the first to regard the UAV platform from an interdisciplinary approach, examining the hardware, software, and operational needs in which the UAV is intended to be deployed, in this case road monitoring [1]. As explained in Section 2.2, such an approach has been often overlooked but nonetheless is much needed to transform UAV monitoring solutions into usable options that benefit society.

## 6. Conclusions

This study presented a UAV monitoring system that combined the advantages of edge computation and remote processing to achieve a monitoring solution that can be used for both urgent and non-urgent monitoring objectives simultaneously, where existing UAV monitoring systems solely consider a single monitoring objective. The results showed that the proposed system is able to achieve real-time data dissemination to road operators on the ground and reinforces the value of UAVs for monitoring complex scenes. The trial conducted near the A-2 highway showed the potential of the system to be applied to real-world scenarios. Future work will aim at optimizing the designed system by considering different design choices in order to obtain faster data transmissions, web application performance, or deep learning deployment, while continuously regarding developments in the field of IoT and edge computation.

**Author Contributions:** Conceptualization, S.T.; methodology, S.T.; software, S.T.; formal analysis, S.T.; writing—original draft preparation, S.T.; writing—review and editing, S.T., F.N., G.V., I.S.d.l.L. and N.K.; visualization, S.T.; supervision, F.N., G.V. and N.K. All authors have read and agreed to the published version of the manuscript.

**Funding:** Financial support has been provided by the Innovation and Networks Executive Agency (INEA) under the powers delegated by the European Commission through the Horizon 2020 program "PANOPTIS—Development of a decision support system for increasing the resilience of transportation infrastructure based on combined use of terrestrial and airborne sensors and advanced modelling tools", Grant Agreement number 769129.

**Data Availability Statement:** Data sharing is not applicable.

**Acknowledgments:** We acknowledge Vertical Technologies (Droneslab B.V.) as the manufacturer of the DeltaQuad Pro and thank them for their support during the development phase. We acknowledge the support of the Spanish Ministry of Transport, Mobility and Urban Agenda in the integration of PANOPTIS technologies into the A2-Highway (Section 2), part of the Spanish Network of first generation highways.

**Conflicts of Interest:** The authors declare no conflict of interest.

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
