# Peer review of "Towards Improved Unmanned Aerial Vehicle Edge Intelligence: A Road Infrastructure Monitoring Case Study"

_remotesensing, doi:10.3390/rs14164008_

Round 1

Reviewer 1 Report

The second version of the manuscript reflects a considerable amount of editing. In doing so, the authors have successfully addressed my own comments on the previous version. Additionally, I believe they have also addressed the comments by the other reviewers. This resulted in an improved presentation and readability for the manuscript and made it more informative for the interested readers. Adding the details, results and analysis for the demo project provided further credibility and relevance.

One minor typo is on Figure 5c where the first cell reads “Semantic segmenta-”. The authors can use a smaller font size or a better abbreviation for the two words.

Author Response

Thank you for the nice words. We have corrected the typo in Figure 5.

Reviewer 2 Report

A Road Infrastructure Monitoring system and some results were introduced in the manuscript, but it is difficult for readers to get the innovation ideas and methods. Some of the Comments and Suggestions for Authors:

1. It was claimed that the designed system can carry out different road monitoring tasks such as on-board object detection or scene segmentation, as well as off-board 3D mapping, and the results showed car detection or scene segmentation, so suggest to give results of 3D mapping.

2. Figure 4 does not present efficent important information for the system design, so it can be deleted.

3. Suggest to give a definition of pipeline.

4. The future works were discussed extensively, but I think it is too long, suggest to simplify Part 5.

Author Response

  1. It was claimed that the designed system can carry out different road monitoring tasks such as on-board object detection or scene segmentation, as well as off-board 3D mapping, and the results showed car detection or scene segmentation, so suggest to give results of 3D mapping.

We politely decline this suggestion for two reasons: 1) We are unfortunately limited to 18 pages, which we have already reached without an in depth discussion of the deep learning and analytical suite. 2) The hardware and software benchmarks presented in section 4.2 are most important to demonstrate the communication and processing pipeline because they represent the foundation of our proposed system. For these reasons we made the decision early on to omit the visual results of the 3D mapping and simply state the processing benchmarks of this task on the Ground Control Station (GCS), see line 584. We hope the reviewer understands.

  1. Figure 4 does not present efficent important information for the system design, so it can be deleted.

In line with our reasoning above regarding what we define as efficient information, we believe that this figure and accompanying text provides important information regarding the communication pipeline and on how the system is operationalized while in the field, and should therefore, remain in the manuscript.

  1. Suggest to give a definition of pipeline.

The definition of pipeline was already given in line 78: “In practice, this means that the processing pipeline, from image-capture to information dissemination to ground-based personnel, needs to be sufficiently quick such that road operators are able to take mitigating actions within seconds.”

  1. The future works were discussed extensively, but I think it is too long, suggest to simplify Part 5.

We have split section 5 into a separate discussion and conclusion section and made smaller edits to improve the flow the new sections. In doing so, we have simplified the discussion regarding future works.

Reviewer 3 Report

 1. Add the objectives of the study in a numbered format.

2. Instead of the superscripts 1,2,3 , i suggest adding the link next to the key text in the manuscript. Please follow MDPI formatting.

3. There are a lot of abbreviations in the paper, the authors should add a table of abbreviations and explain all the acronyms in the table. 

4. Framework and system are interchangeably used in the paper. Please correct them where applicable.

5. As I previously stated, The discussions and conclusion sections must be separated, and each discussed properly. In the discussion section the focus must be on the “remote sensing” aspects and how your proposed system will help enhance the body of remote sensing knowledge.  Support this with relevant references.

Author Response

  1. Add the objectives of the study in a numbered format.

There is only one objective, denoted in line 91. To avoid the appearance of a second objective, we have removed the words “In addition, …” in subsequent line 93. We hope this improves clarity.

  1. Instead of the superscripts 1,2,3 , i suggest adding the link next to the key text in the manuscript. Please follow MDPI formatting.

We couldn’t find MDPI formatting guidelines for hyperlinks. We have followed your suggestion and added them next to the text.

  1. There are a lot of abbreviations in the paper, the authors should add a table of abbreviations and explain all the acronyms in the table.

We politely decline this suggestion for two reasons: First, we are limited to 18 pages, considering this is a technical note, which we need for content. Second, all abbreviations have been defined upon first usage, even the more common ones (GPU, CPU, etc.).

  1. Framework and system are interchangeably used in the paper. Please correct them where applicable.

We have corrected the instances where they were used interchangeably.

  1. As I previously stated, The discussions and conclusion sections must be separated, and each discussed properly. In the discussion section the focus must be on the “remote sensing” aspects and how your proposed system will help enhance the body of remote sensing knowledge. Support this with relevant references.

We have split section 5 into a separate discussion and conclusion section.  Besides edits made to improve the flow of the discussion, we have added a paragraph that explicitly states our contribution to the remote sensing domain while citing relevant references. See line 628.

Reviewer 4 Report

The paper introduces the development of a UAV agnostic system developed for multiple road infrastructure monitoring tasks in real time. It was developed in the context of the H2020 PANOPTIS project, which in my opinion gives value to the work, since it is not only at an early stage of research, but has been tested in a real use case.

The results section shows the clearly advantage of processing the images onboard and then send the labelling result instead of sending the original images and processing them in the GCS.

In my opinion, the paper is well explained and ready to be published. Just some few questions:

11- The developed GCS software can manage multiple UAVs at the same time? Use multiple UAVs can give more value to your solution.

22- Table 1 shows the UAV specifications, where the MTOW is specified, but what is the payload weight? In other words, from the MTOW how much weight is left for the camera, on-board processing unit, transmission system, etc.

33- In section 4.2, line es 574-575, you mention that the high power usage of the on-board processing unit reduce the flight time of the UAV, but you do not specify the flight time using this on-board processing unit. In my opinion this information is very important, since it will limit the use of the system, and may make it not a viable system for real use.

This manuscript is a resubmission of an earlier submission. The following is a list of the peer review reports and author responses from that submission.

Round 1

Reviewer 1 Report

This manuscript presents a design for system that can carry out different road monitoring tasks. The manuscript shows the preliminary results of a successfully working edge-assisted monitoring system for object (vehicle) detection and scene segmentation.

Properly labelled “Technical note”, the manuscript is successful in detailing the hardware setting and interfaces of such system as well proving the system’s efficiency in dealing with the computational strain typically associated with the performing intended tasks.

The manuscript is well-timed as it addresses a current interest and application field (road monitoring by UAV). Furthermore, the manuscript appropriately presents the technical details in well written and structured manner.

Minor typos:

Lines 50-52: the sentence is confusing, as the word “quality” was used twice in conflicting meninges in the same sentence.

Line 67: “manmade or human events” has the same meaning. I guess that the authors meant to say either “manmade or natural events” or “natural and human events”.

Line 268: error in citing Figure 1.

Other suggestions:

Although they are very common, but I suggest that the authors define (spell out) at the first instance the terms CPU, GPU, IoT, GNSS.

It will be interesting to read about the performance of the system over differently colored sections of the road (e.g., Figure 5.a) as this situation is common due to paving the roads by different material or at different times.

Reviewer 2 Report

The manuscript provides an introduction of an road infrastructure monitoring UAV system. Some comments:

  1. The presentation looks like a system introduction, suggest to give more descriptions for the design ideas, new  methods and the technology contributions.
  2. The specification of the system integration and its characteristics were presented, as claimed in the abstract and introduction part, the deep learning algorithm, image processing and communication pipeline design method, multiple task scheduling method and edge offloading model should be the main content of the manuscript. But readers could not reach some detailed information about that.

  3.  The paper can be better organized, for example,  the topic of part 2.1 is UAV-based road infrastructure monitoring, but multiple task and edge computing was summarized in the end of this part.

  4.  

    The system informations were presented in the related work part,  I suggest to  give analysis for their technical method.  

Reviewer 3 Report

Thank you for providing me with the opportunity to read “Towards Improved Unmanned Aerial Vehicle Edge Intelligence: A Road Infrastructure Monitoring Use-Case”. I have the following comments:

  • In my humble opinion, the paper is more suited for the journal “drones”. I don’t see much relevance to remote sensing.
  • The authors need to carefully establish and discuss the connection between UAVs, remote sensing and road monitoring in the introduction section.
  • There seems to be some confusion between the understanding of the terms “remote sensing” and “remote processing”. The paper is based on remote processing and is hence more suitable for UAVs/computing journals. I recommend trying “drones”. The link in the current paper with remote sensing can not be clearly seen.
  • UAVs have been used by multiple studies for road monitoring. Therefore, the authors need to clearly discuss the key innovations of their studies compared to these recently published papers. For example, see the following (the list is not exhaustive):
    1. Elloumi, M., Dhaou, R., Escrig, B., Idoudi, H., & Saidane, L. A. (2018, April). Monitoring road traffic with a UAV-based system. In 2018 IEEE Wireless Communications and Networking Conference (WCNC) (pp. 1-6). IEEE.
    2. Huang, H., Savkin, A. V., & Huang, C. (2021). Decentralized autonomous navigation of a UAV network for road traffic monitoring. IEEE Transactions on Aerospace and Electronic Systems, 57(4), 2558-2564.
    3. Roberts, R., Inzerillo, L., & Di Mino, G. (2020). Using UAV based 3D modelling to provide smart monitoring of road pavement conditions. Information, 11(12), 568.
    4. Wang, X., OuYang, C., Shao, X., & Xu, H. (2021, February). A method for UAV monitoring road conditions in dangerous environment. In Journal of Physics: Conference Series (Vol. 1792, No. 1, p. 012050). IOP Publishing.
    5. Yang, J., Zhang, J., Ye, F., & Cheng, X. (2019, May). A UAV based multi-object detection scheme to enhance road condition monitoring and control for future smart transportation. In International Conference on Artificial Intelligence for Communications and Networks (pp. 270-282). Springer, Cham.
  • The authors should use MDPI format for referencing. Please follow the template.
  • Please correct the errors in the manuscript, such as “Error! Reference source not found.” Line 268.
  • Please add a method flow chart at the start of the method portion and also provide an overview of the method subsection in a few lines.
  • A comparison of the UAV used by the authors with the UAVs used in similar studies is needed to justify how this UAV is better.
  • The results section must be related to remote sensing and duly discussed in terms of the practical application for remote sensing purposes. Please revise and add detailed paragraphs to discuss the applicability in relation to remote sensing.
  • The discussion section isn’t really a discussion, it reads more like a conclusion. Please add a proper discussion section and focus on the “remote sensing” aspects and how your proposed system will help enhance the body of remote sensing knowledge.
  • A proper conclusion section is needed with the limitations of the experiment and how this work can be enhanced in future.